# Antibacterial and Antibiofilm Activities of β-Lapachone by Modulating the Catalase Enzyme

**DOI:** 10.3390/antibiotics12030576

**Published:** 2023-03-14

**Authors:** Mushtaq Ahmad Mir, Somaya Ahmed Altuhami, Sukanta Mondal, Nasreena Bashir, Ayed A. Dera, Mohammad A. Alfhili

**Affiliations:** 1Department of Clinical Laboratory Sciences, College of Applied Medical Sciences, King Khalid University, Abha 61421, Saudi Arabia; 2Department of Biological Sciences, Birla Institute of Technology and Science, Pilani, K.K, Birla Goa Campus, Zuarinagar 403726, Goa, India; 3Medical and Molecular Genetics Research, Department of Clinical Laboratory Sciences, College of Applied Medical Sciences, King Saud University, Riyadh 12372, Saudi Arabia

**Keywords:** β-Lapachone, bactericidal, biofilms, antibiotic resistance, zone of inhibition, antimicrobial activity

## Abstract

Background: Bacterial infections constantly have a large impact on public health, because of increased rates of resistance and reduced frequency of development of novel antibiotics. The utility of conventional antibiotics for treating bacterial infections has become increasingly challenging. The aim of the study was to assess the antibacterial effect of β-Lapachone (β-Lap), a novel synthetic compound. Methods: The antibacterial activity of the β-Lap compound was examined against laboratory strains by agar well diffusion method and broth dilution assay. Growth kinetics in presence of β-Lap on *Staphylococcus aureus*, *Staphylococcus epidermidis*, and *Pseudomonas aeruginosa* (ATCC 27853) were assessed by microplate alamarBlue assay. Crystal violet blue assay was used for biofilm inhibition and biofilm eradication. *P. aeruginosa* catalase (KatA) complexed with β-Lap was modeled using molecular docking approach. Results: β-Lap exhibited potent antimicrobial activity against laboratory strains of bacteria with MIC of 0.2 mM for *S. saprophyticus* and *Staphylococcus aureus*, and 0.04 mM for *Staphylococcus epidermidis* and *Pseudomonas aeruginosa* ATCC 27853. The inhibition of catalase enzyme was found to be the cause for its antibacterial activity. Bioinformatics analysis suggests that β-Lap can inhibit KatA activity by interacting with catalase proximal active site and heme binding site. The activity of some commercial antibiotics was enhanced in association with β-Lap. In addition, β-Lap inhibited the biofilm formation and eradicated the already formed and ultra-mature biofilms of aforesaid bacterial strains. Conclusion: These observations indicated that β-Lap could be a promising antibacterial agent for the treatment and prevention of infectious diseases.

## 1. Introduction

Natural products are important sources of new drug discoveries and play an important role in drug development [1]. Synthetic molecules and natural products could practically resolve the long-standing medication design challenges. The development of active antimicrobials from natural products would be aided by modern chemistry and bioinformatics approaches [2]. 35% of medications available in the market are reported to be derived from natural materials, with 25% of those accounted for by plants [3]. Natural products, particularly antibiotics and therapeutic agents of cancer, with their distinctive benefits, are essential in the process of novel drug discovery [4].

However, bacterial infections have a significant impact on public health. Treatment of bacterial infections is becoming more difficult as a result of an increase in the number of patients with complex underlying problems and microorganisms resistant to current antimicrobial medications [5,6,7]. These issues, as well as drug cytotoxicity and a limited therapeutic spectrum, have prompted the development of new antimicrobial drugs [5]. One of the main reasons to discover new antimicrobial agents or to improve the efficacy of already available antibiotics is antimicrobial resistance (AMR), which is one of the world’s most serious public health issues at present. Worldwide antibiotic resistance threatens our progress in healthcare, food production, and ultimately life expectancy. Therefore, the identification of new antimicrobial agents is of immense need to save lives and protect health systems.

β-Lap is a natural compound isolated from the heartwood of lapacho tree (*Tabebuia impetiginosa*) found in different parts of South America [8]. Numerous studies have demonstrated that β-lap has a variety of pharmacological actions, including anticancer [9], anti-*Trypanosoma cruzi*—the causative agent of Chagas disease—[10], antibacterial [11], and antifungal properties. Having antifungal activity against azole-resistant *Candida* spp. isolates not only inhibits the biofilm formation but disrupts the already formed mature biofilms. Cristian Salas et al. used β-Lap to treat Chagas disease, one of the most serious endemic disorders caused by *Trypanosoma cruzi*, and found that β-Lap inhibited epimastigote development in *Trypanosoma cruzi* cells and trypomastigote vitality in vitro [10].

β-Lap possesses antibacterial properties which act against a variety of bacteria species, including *M. tuberculosis*, its multidrug resistant (MDR) clinical isolates and methicillin resistant *S. aureus* (MRSA). β-Lap showed additive effect in combination with first line tuberculosis drugs and synergic effect with N-acetylcysteine against *M. tuberculosis* H37Rv [12]. In addition, it synergistically inhibited the growth of methicillin resistant *S. aureus* (MRSA) in combination with fluoroquinolones, carbapenems, and other beta-lactam antibiotic [13]. Generation of hydrogen peroxide and superoxide anions in bacterial cells has been demonstrated as one of the mechanisms for the action of β-Lap [14].

Nutrient limitation, slow growth, adaptive stress responses, and formation of persistent cells are hypothesized to constitute a multi-layered biofilm, which leads to the poor antibiotic penetration [15] and thus difficult eradication of biofilm producers [16]. It is believed that the reduced antibiotic susceptibility of bacteria in biofilms is due to a combination of poor antibiotic penetration, an altered microenvironment, adaptive responses, and the presence of bacterial persisted cells. Therefore, there is a need to develop a novel antibiotic to inhibit the biofilm formation and to disrupt the formed biofilms, which would allow the cells to grow actively and thus might help the existing antibiotics to clear infections involving biofilms which are refractory to current treatments [17].

## 2. Materials and Methods

### 2.1. Solvents, Media, and Chemicals Used in This Study

β-Lap was purchased from Solarbio Life Science (Beijing, China) and dissolved in dimethyl sulfoxide (DMSO). Gram’s crystal violet, a bacteriological stain, was obtained from TITAN BIOTECH. Phosphate buffered saline (PBS) tablets were obtained from Sigma Aldrich^®^, Dorset, UK. Methanol (≥99.8%) was purchased from Sigma-Aldrich^®^ GmbH Chemie, Steinheim, France. Acetic acid (≥99.8%) was purchased from Sigma-Aldrich^®^ GmbH Chemie, Steinheim, Germany. Six antibiotics belonging to different classes and potencies were purchased from different pharmaceutical companies. Gentamicin (10 µg) from Becton Dickinson, NJ, USA, vancomycin (30 µg), amikacin (30 µg), tetracycline (30 µg), ciprofloxacin (5 µg) from Bioanalyse^®^ Ankara, Turkey, and amoxicillin (30 µg) from Oxoid™ Basingstoke, Hampshire, UK.

### 2.2. Bacterial Strains Used for Antimicrobial Susceptibility Testing

Laboratory bacterial strains of *Staphylococcus epidermidis*, *Pseudomonas aeruginosa* ATCC 27853, *Staphylococcus saprophyticus*, *Staphylococcus aureus*, *Pseudomonas* spp., *Shigella*, *Escherichia coli*, *Klebsiella pneumonia*, *Streptococcus mutans*, *Mycobacterium smegmatis*, and *Enterococcus faecalis* grown in trypticase soy broth were tested for antibacterial activity of β-Lap. All strains are preserved as glycerol stocks in the freezer at −80 °C.

### 2.3. Zone of Inhibition

Agar well diffusion method is widely used to evaluate the antimicrobial activity of plants or microbial extracts. Wells of 6 mm diameter were punched aseptically with a sterile tip on trypticase soy agar (TSA) plates. Using aseptic techniques, inoculums of 0.5 McFarland were prepared by taking a colony from TSA plate into 2 mL of normal saline and mixing it thoroughly by vortexing. Subsequently, the bacterial suspension was diluted 100-fold by transferring 20 µL of the bacterial suspension in to 2 mL of sterile normal saline. Above diluted bacterial suspensions were spread over the agar plate’s surface using a sterile cotton swab to obtain a uniform microbial growth on plates. Then, 20 µL of 2.5 mM β-Lap dissolved in DMSO was dispensed into the wells of the plate, while, for control, 20 µL of DMSO was dispensed into separate wells. After 24-h incubation, a ruler was used to measure the diameter of a clear zone around the well.

### 2.4. Minimum Inhibitory Concentration (MIC) by Broth Dilution Method

MIC, the lowest concentration of antimicrobial agent which completely inhibits the growth of microorganism, was determined by broth dilution method as described [18] with slight modification. In brief, some individual colonies of a bacterial strain were suspended into 2 mL of saline to an optical density of 0.5 McFarland. The culture was subsequently diluted 100-fold in trypticase soy broth. Then, 180 µL of diluted culture was dispensed into the wells of 96-well plate of having periphery wells filled with 200 µL of distilled water to avoid evaporation during incubation period. The 11th column was filled with 200 µL of sterile trypticase soy broth for the control of sterility of growth medium. Wells of the column 2 were reserved for control to which 20 µL of DMSO was added. The 20 μL of stock solutions of 10 mM, 5 mM, 2 mM, 0.4 mM, 0.08 mM, 0.016 mM, 0.0032 mM, and 0.00064 prepared in DMSO were added to the wells of a row from 3rd column to 10th column. For each bacterial strain, three wells per dilution of β-Lap were used. The plates were incubated for 24 h at 37 °C. The lowest concentration of the compound at which no visible growth was observed is considered as MIC for the bacterial strain tested.

### 2.5. Minimum Bactericidal Concentration by Streaking Method

The minimum bactericidal concentration (MBC) is the lowest concentration of an antimicrobial at which microorganism is completely killed. A sterilized loop was used to streak the cultures from the wells of MIC up to the wells of highest concentration of β-Lap on TSA plates. The concentration of β-Lap where no growth was detected is considered as MBC for the bacterial strain tested.

### 2.6. Microplate AlamarBlue Assay (MABA)

The Microplate AlamarBlue^®^ assay (MABA) is a sensitive, rapid, and nonradiometric method, which evaluates cellular health and offers the potential for screening large numbers of antimicrobial compounds. MABA was performed as described [18] with a slight modification In brief, 180 µL of aforementioned 100-fold diluted culture of a bacterial strain in trypticase soy broth containing 1× alamarBlue were dispensed into the wells of 96-well plate. Then, 20 µL of β-Lap at its MIC and 2×MIC were dispensed in triplicate culture wells. For positive control and medium sterility 20 µL of DMSO were added to triplicate wells of culture and growth media, respectively. The periphery wells were filled with 200 µL of sterile water to avoid evaporation of the well content. The plate was sealed with breathable sealing film (Merck) to allow exchange of gasses. The plate was incubated in FLUOstar^®^ Omega microplate reader overnight. The fluorescent readings (excitation/emission maxima at 544/590 nm) were recorded after every 30 min. The average values of blank corrected fluorescence units were plotted against time using Microsoft Excel software. Standard deviations were calculated, and graphs were constructed by using Microsoft Excel software, version 16.66.1.

### 2.7. Antibiotic Activity Assay

To investigate the effect of the β-Lap compound for several commercial antibiotics, the 0.5 McFarland of bacterial suspension was diluted 100-fold in normal saline and the resulting bacterial suspension was streaked on TSA plates using a sterile cotton swab to achieve a uniform spread of microbial colonies. The antibiotic discs (diameter of 6 mm) were placed on the surface of inoculated plate to determine its inhibition zone size. For the effect on commercial antibiotics, 20 µL of β-Lap from 2.5 mM working stock were dispensed onto the antibiotic discs. The size of inhibition zone was measured in millimeters after 18–24 h of incubation at 37 °C.

### 2.8. Crystal Violet Biofilm Assay

First, 180 µL of 100-fold diluted bacterial suspension obtained from 0.5 McFarland were dispensed into the wells of 96-well plate. To determine the minimum biofilm inhibition concentration (MBIC), 20 µL of β-Lap compound at 5-fold dilutions ranging from 0.5 mM to 0.0016 mM were dispensed in triplicate wells. For the control 20 µL of DMSO were dispensed into the triplicate wells. After 24 h of incubation under static conditions of growth at 37 °C, the wells were rinsed thrice with sterile PBS (phosphate-buffered saline, pH 7.4) to remove the non-adherent bacteria. The cells on the walls of wells were fixed with 99% (*v*/*v*) methanol for 15 min. After discarding methanol wells were dried in laminar flow for 5 min. The attached biofilm cells were stained with 0.5% crystal violet (Sigma-Aldrich) for 5 min at room temperature. The excess stain was removed by washing thrice with distilled water and the crystal violet-bound cells were solubilized in 33% acetic acid. The absorbance readings at 570 nm were taken in FLUOstar^®^ Omega microplate reader and the averages of the blank corrected values were plotted using Microsoft Excel. For mature biofilm formation, the cultures were allowed to grow statically for 24 or 48 h at 37 °C. The biofilms formed were washed thrice with PBS and incubated with sterile growth medium containing β-Lap. 24 h later the biofilms were washed, stained, and estimated as described above. The percentage of biofilm inhibition was calculated by the following formula: Biofilm inhibition (%) = [(Control OD570 nm − Treated OD570 nm)/Control OD570 nm] × 100.

### 2.9. Colony Forming Units (CFU)

CFU of the cultures treated with DMSO or β-Lap were enumerated by plating 100 µL of 10-fold dilutions on nutrient agar plates. The colony count was calculated by the formula, CFU/mL = (Number of colonies in 0.1 mL of culture plated/0.1) × dilution factor. For hydrogen peroxide sensitivity assay, cells treated overnight with 1/5xMIC were pelleted down, washed thrice with 1xPBS, and re-suspended in 1xPBS containing 20 mM H_2_O_2_. After one hour of incubation at 37 °C, the cell suspensions were serially diluted and 100 µL were plated on nutrient agar plates. CFUs/mL ere enumerated from DMSO treated and β-Lap treated samples for each strain.

### 2.10. Catalase Activity Assay

For catalase activity, the bacterial cultures containing β-Lap at its 1/100xMIC were incubated overnight in a shaking incubator at 200 RPM. For control, bacterial cultures in presence of DMSO were incubated under similar conditions. The 200 µL of 3% H_2_O_2_ were dispensed into one ml of the overnight culture in a glass tube and mixed immediately. The height of air bubble column formed above the culture was measured using a ruler. The experiment was setup in triplicates for each strain and the average length of the air bubble column was plotted using Microsoft Excel.

### 2.11. Molecular Modeling

Modeled *Pseudomonas aeruginosa* catalase complexed with β-Lapachone using molecular docking approach. Coordinates of crystal structure of *P. aeruginosa* catalase (KatA tetramer, PDB code: 4e37, resolution 2.53 Å) with heme and NADPH bound was downloaded from Protein Data Bank (https://www.rcsb.org/structure/4E37) on 20 December 2022. Bound ligands and water molecules were removed for docking studies. β-Lap structure (.sdf file, PubChem CID: 3885) were downloaded from PubChem database [19] (https://pubchem.ncbi.nlm.nih.gov/compound/3885, accessed on 20 December 2022). To generate KatA-ligand complexes CB-Dock2 [20], cavity guided blind docking, web server available at https://cadd.labshare.cn/cb-dock2/ (accessed on 20 December 2022) was used. Modeled KatA-ligand complexes were analyzed using BIOVIA Discovery Studio Visualizer, v21, 2021, Protein-Ligand Interaction Profiler [21] available at https://plip-tool.biotec.tu-dresden.de/plip-web/plip/index (accessed on 20 December 2022) and PyMOL available at https://pymol.org (accessed on 20 December 2022) to identify biologically relevant KatA-ligand complexes. 

### 2.12. Statistical Analysis

IBM SPSS statistical software version 21 was used to perform the statistical analysis of data. All the data sets were obtained in triplicates from more than one biological replicate and the values were interpreted as means ± standard deviation (SD). One way ANOVA with Tukey’s post-hoc test was used to assess the differences between treated and untreated samples. The *p* < 0.05 was considered statistically significant.

## 3. Results

### 3.1. Antibacterial Activity of β-Lap

Eleven bacterial strains including nine laboratory strains and two reference strains viz *Pseudomonas aeruginosa* (ATCC 27853), *Streptococcus mutans* (ATCC 25175), *Staphylococcus saprophyticus*, *Staphylococcus aureus*, *Staphylococcus epidermidis*,* Pseudomonas* spp., *Shigella*, *Escherichia coli*, *Klebsiella pneumonia*, *Mycobacterium smegmatis*, and *Enterococcus faecalis* were screened for susceptibility to β-Lap by well-diffusion method. It was found that out of eleven bacterial strains tested only four, *S. epidermidis*,* S. aureus*,* S. saprophyticus*, and *P. aeruginosa* (ATCC 27853) were susceptible to β-Lap. A clear zone of growth inhibition (17.7 ± 1.7 mm to 26.7 ± 1.2 mm in diameter) was observed against the susceptible strains (Figure 1 and Table 1). In order to determine the minimum concentration of β-Lap to inhibit the growth of aforementioned bacterial strains, the strains were treated with five-fold dilutions of β-Lap of final concentrations ranging from 1 mM to 0.00064 mM. The MIC was found to be 0.2 mM for *S. saprophyticus* and *S. aureus* and 0.04 mM for *S. epidermidis* and *P. aeruginosa* (ATCC 27853). As expected, the MBC was higher than MIC (Table 1). These results suggested that *S. epidermidis* and *P. aeruginosa* were comparatively more susceptible to β-Lap than *S. saprophyticus* and *S. aureus*.

To further confirm the bactericidal effect of the β-Lap on these strains, CFUs were enumerated after bacterial cells were treated individually with four concentrations (½ MIC, MIC, 2×MIC, and 4×MIC) of β-Lap. For control, bacterial strains were treated with DMSO under similar conditions. It was evident that the growth of bacterial strains decreased in a dose dependent manner of β-Lap (Appendix A). These results suggested that the β-Lap is a potent antimicrobial against all the four bacterial strains tested.

### 3.2. Growth Kinetics of Bacterial Strains in Presence of β-Lap

MABA, an adequate method for testing antimicrobial activity, was used to monitor the growth of β-Lap susceptible bacterial strains in continued presence of different concentrations of β-Lap. AlamarBlue, a growth indicator, is a fluorescent dye that changes its colour from blue to pink by the reduced environment of the growing cells. The 0.5 McFarland bacterial cultures diluted hundred times in trypticase soy broth containing 1x alamarBlue were incubated with β-Lap at its MIC and 2xMIC. For control, the cultures were incubated with DMSO only. Growth media containing 1x alamarBlue was used as a blank for its sterility. The growth at regular intervals of 30 min was monitored in FLUOstar^®^ Omega microplate reader. Blank corrected average fluorescence units (Ex_560_/Em_590_) plotted against the time for each strain (Figure 2) showed that all the four strains tested had a normal pattern of growth in presence of DMSO. However, a decrease in growth was observed for all the three strains tested with the corresponding increase in the concentration of compound β-Lap from MIC to 2x MIC. These results suggest that the growth inhibitory effect is a dose dependent.

### 3.3. Antibacterial Activity of β-Lap in Combination with Commercial Antibiotics

To determine the effect of the β-Lap on antibacterial activity of commercial antibiotics against *S. saprophyticus*, *S. aureus*, *S. epidermidis*, and *P. aeruginosa*, 20 µL of 2.5 mM β-Lap was added to the discs saturated with antibiotics and subsequently placed on agar plates already streaked uniformly with a bacterial culture. The zone of inhibition measured after 24 h (Table 2) indicated that the antibacterial activity of tetracycline, amoxicillin, and vancomycin in combination with β-Lap increased against *S. epidermidis* (zone size 8 mm > antibiotic alone). The case was the same for ciprofloxacin and amoxicillin against *P. aeruginosa* and *S. saprophyticus*, respectively. Like the antibacterial activity of tetracycline against *S. aureus,* the antibacterial activity of gentamycin, tetracycline ciprofloxacin and amoxicillin in association with β-Lap was moderate against *S. saprophyticus* (zone size 5 mm > antibiotic alone). In other settings, the combination of β-Lap with the antibiotics either mildly enhanced their antibacterial activity (zone size 2–4 mm > antibiotic alone) or had neither additive nor indifferent effect against the bacterial strains tested (Table 2).

### 3.4. β-Lap Treated Cells Are Sensitive to H_2_O_2_ Treatment

Since all the four susceptible bacterial strains are aerobic, we assume that the oxidative stress generated by the β-Lap might be the cause for its antibacterial activity. To investigate this assumption, the β-Lap treated cells were subjected to H_2_O_2_ treatment. For the control, DMSO treated cells were subjected to the same treatment under similar conditions. CFUs enumerated showed that the β-Lap treated cells were very sensitive to H_2_O_2_ stress (Figure 3). These results suggest that β-Lap treatment makes cells susceptible to oxidative stress. Since all pathogens have mechanism in place to survive under oxidative stress of macrophages in human host, the catalase of the aerobic bacteria is one of the components of the defensive mechanism to combat the host’s stress. Therefore, we performed another experiment to determine whether the catalase activity of the bacterial strains tested is affected. It was found that the catalase activity of β-Lap treated cells was significantly reduced compared to DMSO treated control cells (Figure 4, Appendix A). More interestingly, we noticed that β-Lap exhibited a very strong antibacterial activity in shaking condition. The possible reason for reduced catalase activity of β-Lap treated cells could be either inhibition of catalase or reduction in its level, which needs further investigation by biochemical and molecular approaches. However, bioinformatic analysis predicted that β-Lap might inhibit the catalase by binding to it directly. Using bioinformatic approach, modeled KatA-ligand interactions (Figure 5) were analyzed and compared with the crystal structure of *P. aeruginosa* catalase complexed with heme and NADPH (Figure 6). Biologically relevant KatA-ligand complexes shown in Figure 5 and the details tabulated in Table 3 suggest that β-Lap can inhibit *P. aeruginosa* catalase KatA activity by interacting with catalase proximal active site and heme binding site.

### 3.5. β-Lap Inhibits Biofilm Formation

To investigate the effect of β-Lap on the biofilm formation of *S. saprophyticus*, *S. aureus*, *S. epidermidis*, and *P. aeruginosa*, the cells were incubated with five individual concentrations (0.5 mM, 0.2 mM, 0.04 mM, 0.008 mM, and 0.0016 mM) of β-Lap for 24 h at 37 °C. For control, the cells were treated with DMSO under similar conditions. The biofilms were developed by crystal violet method and quantified by measuring absorbance at 570 nm. It is clear from Figure 7 that the biofilm formation of the strains inhibited in a dose dependent manner (Appendix A). A significant inhibition of biofilm formation was observed in *S. saprophyticus* (50%) and *S. aureus* (75%) at 0.04 mM concentration of β-Lap, without effecting their growth. Inhibition in growth was observed at concentration higher than 0.04 mM. Therefore, 0.04 mM was considered as MBIC for the above-mentioned strains, as an ideal anti-biofilm agent should not affect the growth of bacteria. Similarly, 0.008 mM concentration of β-Lap was fixed as MBIC for *S. epidermidis*, on which 78% inhibition in biofilm formation was observed. However, β-Lap exhibited minimal inhibitory effect (less than 50%) on biofilm formation of *P. aeruginosa*. MBIC can be defined as the minimum concentration of β-Lap that exhibits highest biofilm inhibition without affecting growth. Growth of all strains measured by absorbance readings at 600 nm suggests that the cells were metabolically active at sub-MBIC concentrations. These results suggest that the compound has promising anti-biofilm properties.

We further explored whether β-Lap disrupts the mature biofilms of these strains. For biofilm disrupting ability, the bacterial cultures were grown for 24 and 48 h to allow the bacterial cells to form mature biofilms. Free planktonic cells were washed off and the biofilms were treated further for 24 h with MBIC and 2xMBIC of β-Lap in the same growth media. The remaining biofilms, which were resistant to β-Lap under given conditions, were quantified. It is clear from Figure 8 that both the mature and extra mature biofilms were disrupted by β-Lap. Among the bacterial strains tested, the mature and ultra-mature biofilms of *S. saprophyticus* and *S. aureus* were highly sensitive to β-Lap. Contrarily, the mature and ultra-mature biofilms of *P. aeruginosa* were least susceptible to β-Lap. Interestingly, the ultra-mature biofilms of *S. epidermidis* showed highest sensitivity to β-Lap. However, it is worth noting that the biofilm inhibition activity of the compound was more prominent than its biofilm disruption capability.

## 4. Discussion

In recent decades the increase in infectious cases caused by bacterial species and the resulting disproportionate use of antimicrobials has led to the development of resistance to antimicrobial drugs [23]. To overcome such hurdles, scientists gained interest in recent decades in identifying new antimicrobials, for which plants and their secondary metabolites are a prospective source of bioactive molecules as potential therapeutic agents [24]. Naphthoquinones, a group of secondary metabolites commonly found in higher plants, have great structural diversity and biological activity.

In the present study, we evaluated the ability of β-Lap, a naphthoquinone, to inhibit the growth of several pathogenic bacteria. Out of eleven bacterial strains screened, only four *S. saprophyticus*,* S. aureus*, *S. epidermidis*, and *P. aeruginosa* were susceptible to the tested concentration (2.5 mM) of β-Lap with zone of inhibition ranging from 17.7 mm to 26.7 mm (Table 1). Among them *S. epidermidis*, *S. saprophyticus*, and *P. aeruginosa* showed the highest zone of inhibition, while *S. aureus* comparatively showed lower zone of inhibition. As expected, a positive correlation was observed between inhibition zone sizes and corresponding MIC values. To further evaluate the bactericidal activity of β-Lap, cells were treated with increasing concentration of β-Lap (1/2xMIC, MIC, 2xMIC and 4xMIC) and as expected CFUs enumerated decreased in a dose dependent matter, suggesting that the β-Lap compound was bactericidal in nature (Appendix A). In addition to this endpoint analysis, growth kinetics of bacterial strains in presence of MIC and 2xMIC of β-Lap confirmed the decrease in growth rate in a dose-depended manner (Figure 2). Though antibacterial activity of the β-Lap has not been studied in detail, our data were consistent with the other previous observations. In one of the studies, β-Lap encapsulated in liposomes inhibited different MRSA isolates [11]. In another study it inhibited the methicillin sensitive and resistant strains of *S. aureus* and methicillin resistant strains of *S. epidermidis* and *S. haemolyticus* [25]. Among the naphthoquinones (lapachol, α-lapachone, and β-lapachone) tested against several strains of MRSA, β-Lap exhibited not only antibacterial activity but synergic activity in association with commercial antibiotics [13]. Since in the present study all the β-Lap susceptible strains are aerobes and Gram positive, we suspected that the activity of catalases might be compromised in such strains. In support of this notion, we assayed the sensitivity of β-Lap treated cells towards H_2_O_2_ treatment. It is evident from the Figure 3 and Figure 4 that the cells treated with β-Lap were very sensitive to H_2_O_2_ treatment due to the large reduction in their catalase activity. The cause of reduction of the catalase activity needs to be investigated, which is our future goal of research. It could be either due to the reduction in the level of catalase or inhibition of its catalytic activity. Molecular modelling predicted the inhibition of catalase activity by binding to the proximal active site and heme binding sites of the enzyme.

The combinational therapy of the new drugs with already existing commercial antimicrobials is an important strategy in the search of more effective antimicrobial agents for treating microbial infections. This can potentially increase efficacy, reduce toxicity, cure faster, prevent the emergence of resistance, and provide a broader spectrum of activity than mono therapy regimens. Combinational effect of β-Lap with other antibiotics on the bacterial strains tested (Table 2) showed that it has a potential to increase the activity of some drugs in vitro. These results are in agreement with the synergistic effects between β-Lap and conventional antimicrobials against methicillin resistant *Staphylococcus aureus* (MRSA) strains [13]. It remains to be seen whether the β-Lap will have the same effect in combination with antibiotics in vivo.

Bacterial cells, being substantially more resistant to drugs and host immune responses in biofilms, pose a great challenge to chemotherapy of the infectious diseases. There is poor antibiotic penetration and slow growth due to nutrient limitation, which makes cells less prone to antibiotics. Hence, there is a need to develop new drugs that could penetrate well and kill the bacterial cells protected by multi-layered extracellular polymeric matrix.

Four bacterial strains *S. saprophyticus*,* S. aureus*,* S. epidermidis*, and *P. aeruginosa* (ATCC 27853) were tested to investigate the effect of β-Lap on their biofilm formation. It was found that the biofilm formation in the tested strains significantly inhibited (50–80% biofilm inhibition) at MBIC, except *P. aeruginosa*. Among the strains *S. saprophyticus*,* S. aureus,* and *S. epidermidis*, the biofilm formation of *S. aureus* and *S. epidermidis* was strongly inhibited. The inhibition was 75% for *S. aureus* and 78% for *S. epidermidis* at their respective MBIC. More importantly, no inhibition of growth occurred in the strains at MBIC (Figure 7), suggesting that β-Lap targets the processes or factors required for biofilm formation without effecting the biological processes required for normal growth of the bacteria. These results suggested that the β-Lap could be potential antibiofilm agent against *S. aureus* and *S. epidermidis.* Not only did β-Lap inhibit biofilm formation, but it disrupted the mature biofilms of these bacteria. It was found that 48-h mature biofilms of *S. epidermidis* were more sensitive to β-Lap as the biofilms were eradicated 11.5-fold by β-Lap treatment than that of control where the biofilms were treated with DMSO alone. Though biofilm formation of *P. aeruginosa* was not inhibited by β-Lap, their 24-h mature biofilms were eradicated 2.7-fold more than that of control, where biofilms were treated with DMSO only. The mechanism behind the inhibition of biofilm formation and eradication of already formed biofilms needs to be investigated. Our results of antibacterial and antibiofilm activities of β-Lap corroborate well with the study of Fernandes AWC, et al., wherein the authors showed that β-Lap and Lapachol Oxime (semi-synthetic derivatives of Lapachol) exhibited antimicrobial and anti-biofilm potential against clinical isolates of *S. aureus* [26].

The results altogether certainly confirm that the β-Lap exhibited strong antibacterial and antibiofilm activities as well as enhanced the efficacy of commercial antibiotics tested. Therefore, β-Lap could be a drug of choice to be used in treating bacterial infections, but its activity in vivo needs to be investigated, which demands further research.

Conclusion: Taken together, our data suggest that β-Lap is a promising novel antibacterial agent that inhibits the bacterial catalase.

## Figures and Tables

**Figure 1 antibiotics-12-00576-f001:**
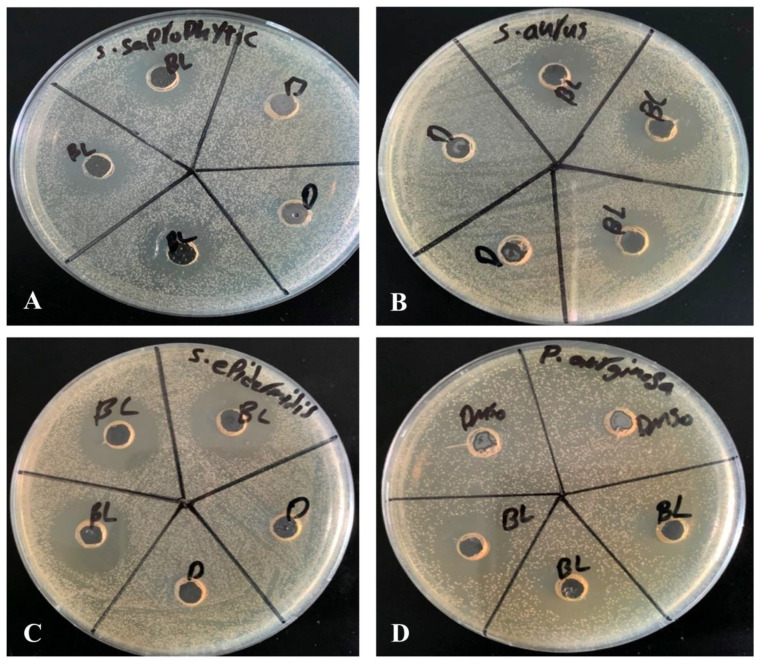
*Zone of inhibition of susceptible strains*: 20 µL each of β-Lap (2.5 mM) and DMSO were dispensed in triplicate and duplicate wells, respectively, of the plates uniformly streaked with bacterial suspension of (**A**) *S. saprophyticus*, (**B**) *S. aureus*, (**C**) *S. epidermidis*, and (**D**) *P. aeruginosa* (ATCC 27853).

**Figure 2 antibiotics-12-00576-f002:**
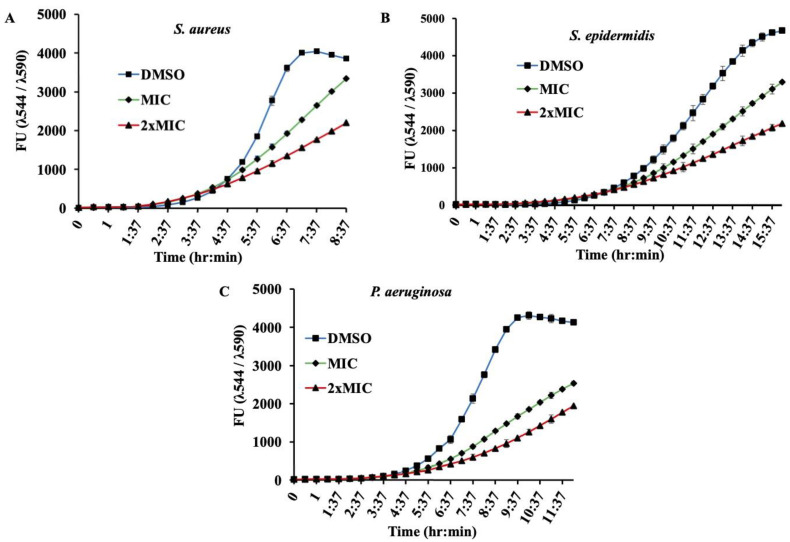
*Growth of bacterial strains in presence of β-Lap*: The growth of (**A**) *S. aureus*, (**B**) *S. epidermidis*, and (**C**) *P. aeruginosa* was monitored in presence of MIC and 2x MIC of β-Lap by recording fluorescence readings at 30 min intervals using alarmBlue as a growth indicator. XX’ shows time in hours and minutes.

**Figure 3 antibiotics-12-00576-f003:**
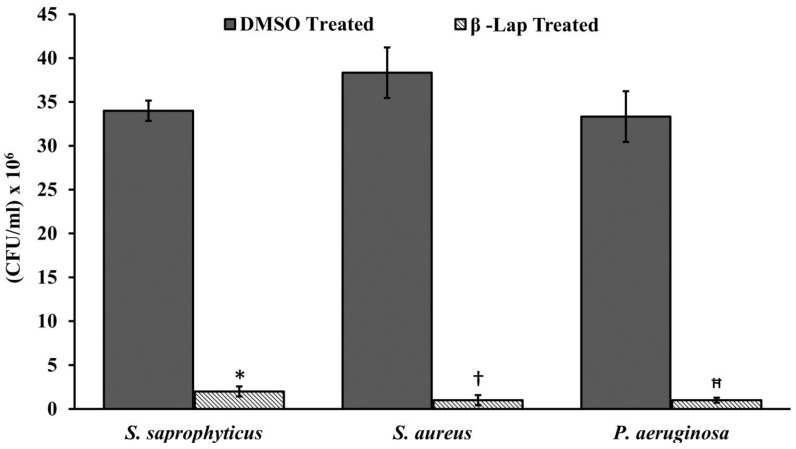
*β-Lap treated cells are sensitive to H_2_O_2_*: Bacterial cultures individually treated with DMSO and β-Lap were subjected to H_2_O_2_ stress. The CFUs enumerated post-H_2_O_2_ stress were plotted. *: *p* < 0.002, †: *p* < 0.004, ꟸ: *p* < 0.007.

**Figure 4 antibiotics-12-00576-f004:**
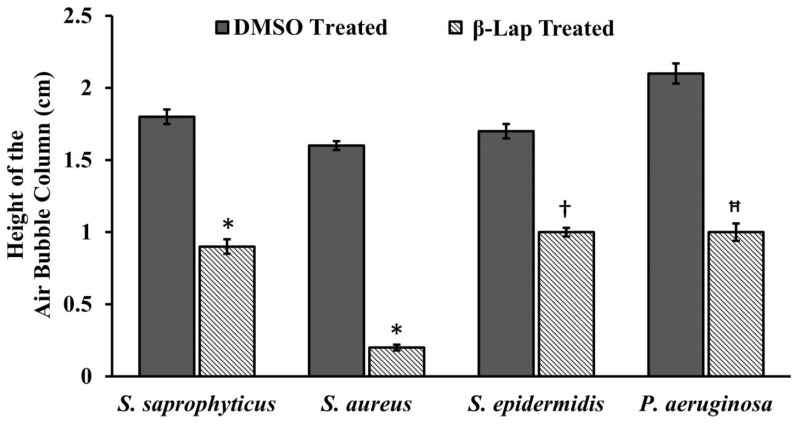
*Catalase inhibition in β-Lap treated cells*: The cells of the bacterial strains individually treated with DMSO (gray columns) and β-Lap (striated columns) were assayed for catalase production by measuring the height of air bubble column after the addition of H_2_O_2_. *: *p* < 0.002, †: *p* < 0.004, ꟸ: *p* < 0.007.

**Figure 5 antibiotics-12-00576-f005:**
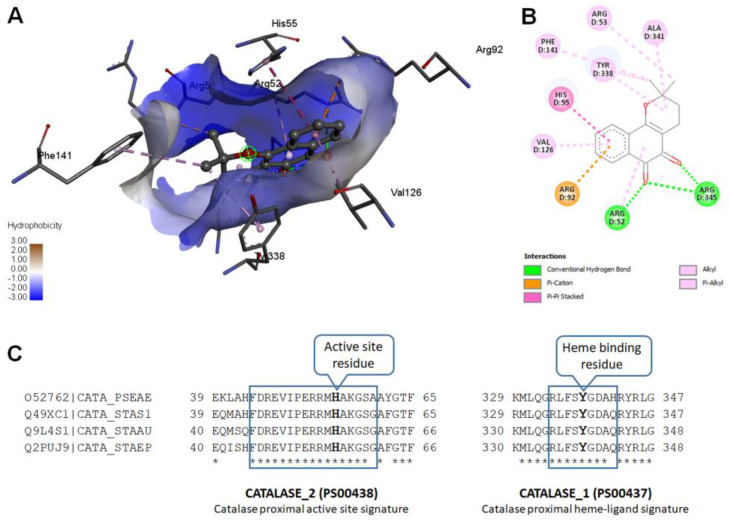
(**A**) Modeled *Pseudomonas aeruginosa* catalase (KatA tetramer, PDB code: 4e37) complexed with β-Lap. For clarity, only β-Lap (ball-and-stick presentation) interaction with one chain (4e37, chain D) of *P. aeruginosa* catalase is shown. (**B**) Modeled non-covalent interactions between β-Lap and KatA. (**C**) Alignments of the conserved ligand binding sites (PROSITE ACs: PS00437 and PS00438) of catalase (KatA) from *Pseudomonas aeruginosa* (UniProtKB AC: O52762), *Staphylococcus saprophyticus* (UniProtKB AC: Q49XC1), *Staphylococcus aureus* (UniProtKB AC: Q9L4S1), and *Staphylococcus epidermidis* (UniProtKB AC: Q2PUJ9). *: conserved residues.

**Figure 6 antibiotics-12-00576-f006:**
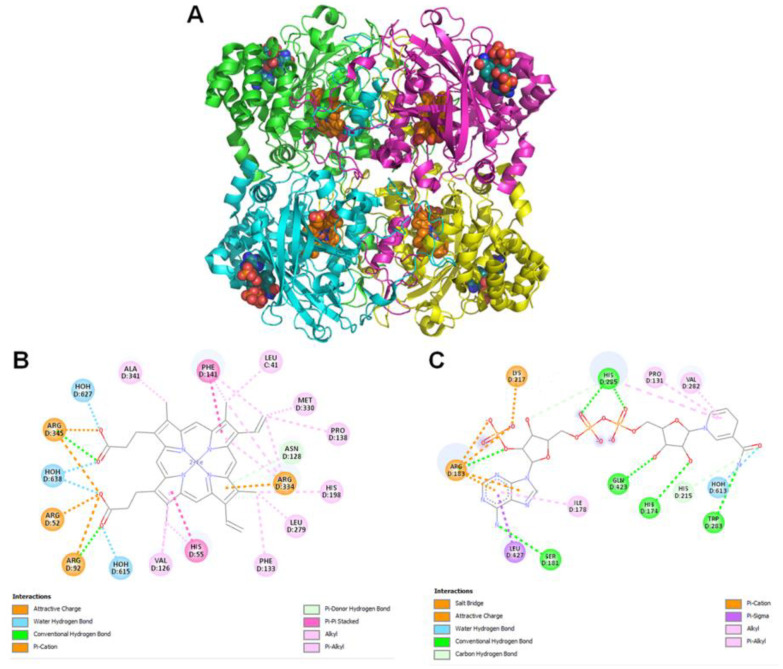
(**A**) Crystal structure of *Pseudomonas aeruginosa* catalase (KatA tetramer, PDB code: 4e37) complexed with heme and NADPH. All four chains are represented as cartoon with different colors and bound ligands presented as spheres (Heme carbons as blue and NADPH carbons as orange). (**B**) Heme and (**C**) NADPH interactions with KatA.

**Figure 7 antibiotics-12-00576-f007:**
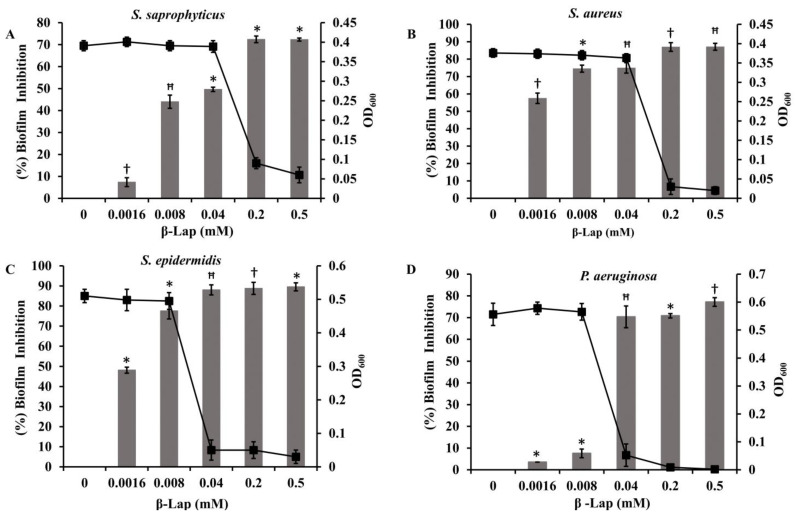
*Inhibition of biofilm formation by β-Lap*: The growth and the biofilms formed by bacterial cultures at different concentration of β-Lap were estimated by crystal violet absorbance at 570 nm and optical density at 600 nm, respectively. The calculation for % inhibition of biofilm formation is discussed in Section 2. The growth curve and bar graph for biofilm inhibition (%) were constructed for each bacterial strain of (**A**) *S. saprophyticus*, (**B**) *S. aureus*, (**C**) *S. epidermidis*, and (**D**) *P. aeruginosa*. Line: Growth curve, Bar: Biofilm inhibition. *: *p* < 0.002, †: *p* < 0.004, ꟸ: *p* < 0.007.

**Figure 8 antibiotics-12-00576-f008:**
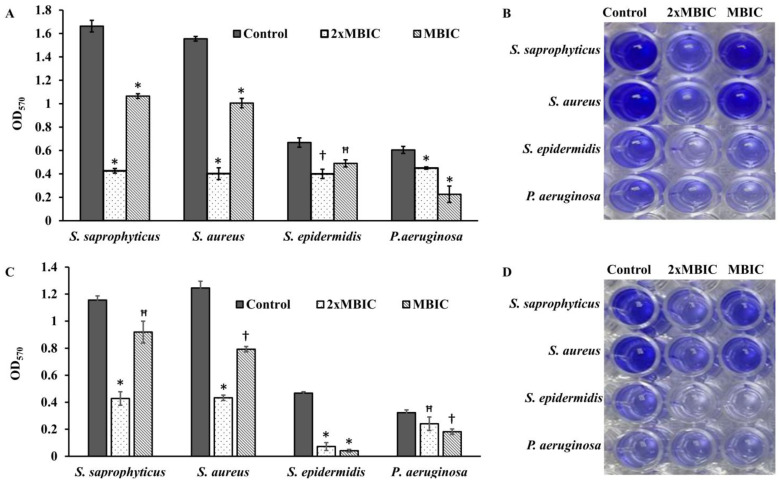
*Mature biofilm disruption activity of β-Lap*: (**A**,**B**) the 24 h mature biofilms were treated with β-Lap at MBIC and 2xMBIC, developed with crystal violet and quantified by absorbance at 570 nm. (**C**,**D**) The 48 h ultra-mature biofilms were developed and quantified under similar conditions. *: *p* < 0.002, †: *p* < 0.004, ꟸ: *p* < 0.007.

**Table 1 antibiotics-12-00576-t001:** Zone of inhibition (mm ± SD), MIC, MBC, and MBIC of β-Lap.

Microorganisms	Zone of Inhibition (mm) ± SD	MIC (mM)	MBC (mM)	MBIC (mM)
*S. saprophyticus*	17.7 ± 1.7	0.20	1	0.04
*S. aureus*	21 ± 0.8	0.20	1	0.04
*S. epidermidis*	26.7 ± 1.2	0.04	0.5	0.08
*P. aeruginosa* (ATCC 27853)	21.7 ± 1.2	0.04	0.5	No effect

**Table 2 antibiotics-12-00576-t002:** Zone of inhibition (mm ± SD) of antibiotics and of their combination with commercial antibiotics.

Antibiotic	*S. saprophyticus*		*S. aureus*		*S. epidermidis*		*P. aeruginosa* ATCC 27853	
	DMSO	β-Lap	DMSO	β-Lap	DMSO	β-Lap	DMSO	β-Lap
Gentamicin	30 ± 0.5	35 ± 0.4	17 ± 0.4	19 ± 0.3	26 ± 0.3	29 ± 0.4	10 ± 0.4	10 ± 0.5
Amikacin	35 ± 0.3	38 ± 0.3	18 ± 0.4	22 ± 0.5	28 ± 0.4	30 ± 0.0	26 ± 0.4	30 ± 0.0
Tetracycline	37 ± 0.3	42 ± 0.0	31 ± 0.0	36 ± 0.3	15 ± 0.5	23 ± 0.0	23 ± 0.5	31 ± 0.4
Ciprofloxacin	32 ± 0.3	37 ± 0.0	32 ± 0.3	34 ± 0.5	35 ± 0.3	37 ± 0.3	R	R
Amoxicillin	16 ± 0.5	21 ± 0.3	9 ± 0.3	17 ± 0.4	17 ± 0.6	25 ± 0.3	11 ± 0.3	11 ± 0.3
Vancomycin	25 ± 0.4	25 ± 0.3	17 ± 0.0	17 ± 0.4	20 ± 0.3	28 ± 0.4	11 ± 0.6	15 ± 0.5

R: Resistant.

**Table 3 antibiotics-12-00576-t003:** Modeled *P. aeruginosa* catalase (KatA tetramer, PDB code: 4e37) interactions with β-Lap.

AutoDock Vina Score (kcal/mol)	Cavity Volume (Å3)	Center (x, y, z)	Docking Size (x, y, z)	Contact Residues *
−10.2	3426	11, 37, 29	19, 34, 19	Chain D: **ARG52** ARG53 **HIS55 ARG92** SER94 SER113 VAL126 **PHE141** ALA312 ALA313 PHE314 SER337 TYR338 **ALA341** HIS342 **ARG345**
−10.1	5757	8, −5, −5	29, 32, 19	Chain A: **ARG52** ARG53 **HIS55 ARG92** SER94 SER113 VAL126 **PHE141** ALA312 ALA313 PHE314 SER337 TYR338 **ALA341** HIS342 **ARG345** *Chain B*: PHE44 ASP45
−10.1	3553	19, 38, 2	27, 32, 28	Chain C: **ARG52** ARG53 **HIS55 ARG92** SER94 SER113 VAL126 **PHE141** ALA312 ALA313 PHE314 SER337 TYR338 **ALA341** HIS342 **ARG345**
−10.1	3420	−1, −5, 23	19, 31, 19	Chain B: **ARG52** ARG53 **HIS55 ARG92** SER94 SER113 VAL126 **PHE141** ALA312 ALA313 PHE314 SER337 TYR338 **ALA341** HIS342 **ARG345**

* Heme binding residues are highlighted in bold, overlapping residues with PS00437 and PS00438 [22] patterns are highlighted with underline.

## Data Availability

All data generated or analysed during this study are included in this article.

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
