# Peer review of "Antibacterial and Antibiofilm Activities of β-Lapachone by Modulating the Catalase Enzyme"

_antibiotics, 2023, doi:10.3390/antibiotics12030576_

Round 1

Reviewer 1 Report

1.  Authors exploited various methods (agar well diffusion method, minimal inhibitory concentration and minimal bactericidal concentration) for assessing the antibacterial effect of β –Lapachone. However, research suggest that media used in agar well diffusion method has not been observed to be that much suitable to calculate the minimum inhibitory concentration (MIC), because it is hard to measure the quantity of the antimicrobial agent that diffused into the agar medium, and certain bacteria do not grow well or at all on it. How does authors justify this method for this investigations?

2.  Additionally, authors are suggesting the elucidation of antibacterial and antibiofilm activities of β –Lapachone through inhibiting the catalase enzyme by using Microplate AlamarBlue assay (MABA).  However, it is well known that the most common and convenient method for measuring catalase activity is the UV spectrophotometric method. Why did authors chose MABA method? If possible, spectrophotometric method may be also evaluated for the catalase activity.

3.   From the result section, the authors suggested that S. epidermidis and P. aeruginosa were comparatively more susceptible to β –Lap than S. saprophyticus and S. aureus. However, it is not fully clear and there is no support from previously published reports incorporated in this manuscript which explains the susceptibility factor. This should be explained.

4.  It is not clear what is the significance of clear zone of growth inhibition. The significance of a clear zone of growth inhibition should be explained. How is the zone of inhibition important regarding the effectiveness of antibiotics? The same may be discussed in the manuscript.

5.  Authors should mention in the manuscript that what makes the bacterial cells substantially more resistant to drugs and host immune responses in biofilms?

6. Growth at regular intervals of 30 min was monitored in FLUOstar Omega microplate reader over the time of 16 hours is shown in Figure 2. However, in this figure units are not clear. If these units (hours : minutes)  indicate other thing, that should be mentioned in the figure caption. Moreover, captions or legends associated with all the figures mentioned in the manuscript are missing. It is difficult to analyze the figures without the figure legends.

7.  As per the (Fig. 7), no inhibition of growth occurred to the strains at MBIC.  What is the reason behind the inhibition of biofilm formation and interruption of mature biofilms of bacteria is in the presence of β–Lap. Elaborate the mechanism of inhibition of biofilm formation from the data?

8.      The manuscript suffers from grammatical and trpographical mistakes. Most notable errors include the placement of definite articles, and awkward sentence structures. The same may be rectified. 

9. In the discussion portion, manuscript is lacking the incorporation of previously published reports, theories and mechanisms with the present outcomes. Making the contract between previously published reports, theories and mechanisms with the present outcomes makes the manuscript scientifically very sound.

10.  There are no references added in the methods portion of manuscript. All references should be added and updated in the manuscript.

Author Response

We are extremely thankful to the reviewer for his efforts and time that he/she spent while going through the manuscript. We appreciate his valuable comments, and we hope that the responses are to his/her satisfaction level.

Reviewer 1

  1. Authors exploited various methods (agar well diffusion method, minimal inhibitory concentration, and minimal bactericidal concentration) for assessing the antibacterial effect of β –Lapachone. However, research suggest that media used in agar well diffusion method has not been observed to be that much suitable to calculate the minimum inhibitory concentration (MIC), because it is hard to measure the quantity of the antimicrobial agent that diffused into the agar medium, and certain bacteria do not grow well or at all on it. How does authors justify this method for this investigations?

Response: TSA has been used in antibacterial activity to assess the zone of inhibition (Miguel Martí, 2018). In our study the bacterial strains grew very well, and zone of inhibition was quite evident. The TSA was used to determine the zone of inhibition and for MIC liquid media was used in 96 well plates.

2.Additionally, authors are suggesting the elucidation of antibacterial and antibiofilm activities of β –Lapachone through inhibiting the catalase enzyme by using Microplate AlamarBlue assay (MABA).  However, it is well known that the most common and convenient method for measuring catalase activity is the UV spectrophotometric method. Why did authors chose MABA method? If possible, spectrophotometric method may be also evaluated for the catalase activity.

Response: we took advantage of catalase test to qualitatively determine whether the catalase is active. The height of the air bubble column formed in a glass tube above the culture was measured to determine the effect of β –Lapachone on catalase. The MABA was used to evaluate the effect of β –Lapachone on the growth curve of bacterial strains. Reviewer is right in his opinion that UV spectrophotometric method could be used to measure the activity of catalase. This is our future aim to determine the Km, Vmax, and mode of inhibition of catalase (competitive, noncompetitive etc) by β –Lapachone. In present study we just found that catalase is one of the targets of β –Lapachone.

  1. From the result section, the authors suggested that S. epidermidis and P. aeruginosa were comparatively more susceptible to β –Lap than S. saprophyticus and S. aureus. However, it is not fully clear and there is no support from previously published reports incorporated in this manuscript which explains the susceptibility factor. This should be explained.

Response: It is the observation of this study as there is not much study done on antibacterial activity, where all these 4 bacterial strains have been tested for susceptibility. There are few reports on antibacterial activity against MRSA. In one such reports the authors showed MIC of lapachone 2-8 ug/ml against clinical and reference strains of S. aureus (Guiraud P, 1994), and in other study the MIC ranged from 2-4mg/ml against MRSA in liposomal formulation (Cavalcanti IM., 2015). Because of varying laboratory conditions, one cannot compare the susceptibility of strains against lapachone investigated under different lab conditions.

  1. It is not clear what is the significance of clear zone of growth inhibition. The significance of a clear zone of growth inhibition should be explained. How is the zone of inhibition important regarding the effectiveness of antibiotics? The same may be discussed in the manuscript.

Response: sometimes satellite colonies come up with time if plates are incubated for longer times. We didn’t notice those colonies coming up when plates were incubated for longer times. The zone of inhibition cannot be used to measure the effectives of antibiotics as it depends on diffusion rate of an antibiotic in agar, which could be a variable factor among antibiotics tested on same plate.

  1. Authors should mention in the manuscript that what makes the bacterial cells substantially more resistant to drugs and host immune responses in biofilms?

Response: Bacteria in biofilms resist antibiotics via several mechanisms, including (i) decreased penetration or diffusion of antimicrobial agents into biofilms, (ii) increased activity of multidrug efflux pumps, (iii) involvement of quorum sensing systems, (iv) starvation or stress responses, and (v) genetic switches that turn on genes involved in biofilm production. Biofilms also hinder the recognition of bacteria by immune system. These statements have been included in manuscript. 

  1. Growth at regular intervals of 30 min was monitored in FLUOstar Omega microplate reader over the time of 16 hours is shown in Figure 2. However, in this figure units are not clear. If these units (hours : minutes)  indicate other thing, that should be mentioned in the figure caption. Moreover, captions or legends associated with all the figures mentioned in the manuscript are missing. It is difficult to analyze the figures without the figure legends.

Response: The statement has been modified in the text as ‘growth at regular intervals of 30 min was monitored in FLUOstar Omega microplate reader’, because for all the strains the time in the figure is not 16 hours. The XX’ shows the time in hours and minutes. It has been mentioned in figure legend also. Sorry for the inconvenience of having legends separate from figures. Legend have now been incorporated below the figures in the manuscript.

  1. As per the (Fig. 7), no inhibition of growth occurred to the strains at MBIC.  What is the reason behind the inhibition of biofilm formation and interruption of mature biofilms of bacteria is in the presence of β–Lap.Elaborate the mechanism of inhibition of biofilm formation from the data?

Response: MBIC is a minimal concentration at which biofilm formation is inhibited without affecting the growth of bacteria. An ideal antibiofilm agent is one which has very minimal or no effect at growth of cells but could inhibit the biofilm formation. One of the reasons for inhibition of biofilm formation or rupture would be reduced production of factors or inhibition of the factors that play role in biofilm formation and their maturation. Lapachone not only inhibits the biofilm formation but also ruptures the already formed biofilms. It is too early to elaborate the mechanism for biofilm inhibition or their eradication. Transcriptome analysis would be ideal to know the set of genes involved, which is our future aim.

8.The manuscript suffers from grammatical and trpographical mistakes. Most notable errors include the placement of definite articles, and awkward sentence structures. The same may be rectified. 

Response: typos have been fixed and manuscript is grammatically improved.

  1. 9. In the discussion portion, manuscript is lacking the incorporation of previously published reports, theories and mechanisms with the present outcomes. Making the contract between previously published reports, theories and mechanisms with the present outcomes makes the manuscript scientifically very sound.

Response: the discussion section has been improved

  1. There are no references added in the methods portion of manuscript. All references should be added and updated in the manuscript.

Response: the references has been added in methods section of MIC, MABA, and molecular modeling.

  1. Miguel Martí, Belén Frígols, and Angel Serrano-Aroca. Antimicrobial Characterization of Advanced Materials for Bioengineering Applications. J Vis Exp. 2018; (138): 57710.doi: 10.3791/57710

  2. Guiraud P, Steiman R, Campos-Takaki GM, Seigle-Murandi F, Simeon de Buochberg Comparison of antibacterial and antifungal activities of lapachol and beta-lapachone. M.Planta Med. 1994 Aug; 60(4):373-4. doi: 10.1055/s-2006-959504.

  1. Cavalcanti IM, Pontes-Neto JG, Kocerginsky PO, Bezerra-Neto AM, Lima JL, Lira-Nogueira MC, Maciel MA, Neves RP, Pimentel MF, Santos-Magalhães NS Antimicrobial activity of β-lapachone encapsulated into liposomes against meticillin-resistant Staphylococcus aureus and Cryptococcus neoformans clinical strains..J Glob Antimicrob Resist. 2015 Jun;3(2):103-108. doi: 10.1016/j.jgar.2015.03.007. Epub 2015 Apr 30. PMID: 27873657

Reviewer 2 Report

Comment to the original paper

The original paper entitled “Antibacterial and antibiofilm activities of β-lapachone by modulating the catalase enzyme” was submitted to Antibiotics. It described the in vitro activity of β-lapachone to inhibit tested bacteria and their biofilms. After considering the paper, I suggest rejection with the following comments.

Overall comment

1. English correction is necessary.

2. No result of figure and table in the paper. No legend of supplement figure.

3. No statistical analysis to compare different samples in each experiment.

In abstract

4. “minimal inhibitory concentration (MICs) and minimal bactericidal concentration (MBCs)”. It can be replaced with broth microdilution assay.

5. Give the full name of MABA

6. Give the abbreviated name after the first full name. Ex. β-lapachone (β-lap) or bacterial names.

7. Some word corrections such as “Growth kinetics inhibitin”, “biofilm inhibition assays, Biofilm eradication assay”.

In introduction

8. Ref 5 may not be suitable for this overall statement “Treatment of bacterial infections are getting more difficult as a result of …”

9. Figure and molecular weight of β-lapachone can help readers to understand the paper.

10. What is its MDR clinical isolates? Full name of MDR.

11. beta lactams can be changed to “other beta-lactam antibiotic”

In method

12. Method 2.2; 1. Italic font of bacterial name. 2. Give the abbreviated name of TS. Ex. trypticase soy (TS) broth.

13. Method 2.4; 1. It is not necessary to give the detail of column number of the microplate. 2. “200 µL of distilled water” Is it sterile water? 3. Did the negative control and samples include 200 µL of sterile water? 4. What is “5 mM stock were prepared in DMSO”?

14. Method 2.5; What is agar used to grow the cells?

15. Method 2.6; 1. The advantage of MABA can be removed. 2. The positive control is 20 mL DMSO + culture (media + cells). Is it negative control? 3. 20 mL DMSO + media is the blank for subtraction. 4. Change “water” to “sterile water”.

16. Method 2.7; Did it have an empty disc containing β-lap?

17. Method 2.8; 1. Full name of MBIC. 2. Unit of 0.0016.

18. Method 2.9; 1. Change the reference of https://doi.org/10.1093/nar/gkac394 and listed it in the reference section.

In result

19. Result 3.1; 1. Italic font of bacterial names. 2. No data of figure and table, I cannot give comment.

20. Result 3.2; 1. There were 4 sensitive strains that β-lap inhibited. Why only three tested strains had a decrease in growth? 2. MIC or 1x MIC was used frequently. The authors could use one pattern for consistency. 3. The compound decreased the bacterial growth. This result should be related to its kinetics.

21. Result 3.3; 1. The amount of compound or 20 mL of 2.5 mM β-lap should replace “2.5 mM β-lap”. 2. What is the effect of the combination of β-lap and antibiotics? (addition or synergy). 3. Docking parameters should be provided. 4. Catalase proximal active site for β-lap should be provided. 5. Fig or Figure, choose one pattern for consistency. 6. No explanation of Supplement Fig 2.

22. Result 3.4; 1. What is “including 0.5 mM of β –Lap”? 2. How to calculate the inhibitory percentage of biofilm? 3. Give the definition of MBIC and why 0.04 mM was MBIC? 4. What is MBIC of P.aeruginosa?

In discussion

23. “Out of ten bacterial strains screened”. Why it was 10 as 11 strains were tested in the method and result.

24. To determine the synergy of β-lap and antibiotics, The checkerboard assay should perform to calculate the fractional inhibitory concentration (FIC) index in order to discuss the consequence of this combination.

25. More discussion on kinetic rate, IC50 on biofilm, and interaction between tested compound and catalase will improve the paper quality.

In conclusion

26. It is overstated that β-lap could inhibit at transcriptional level.

Author Response

We are extremely thankful to the reviewer for his efforts and time that he/she spent while going through the manuscript. The reviewer has keenly reviewed the manuscript. We appreciate his/her valuable comments, suggestions, and we hope that the responses are to his/her satisfaction level.

Comment to the original paper

The original paper entitled “Antibacterial and antibiofilm activities of β-lapachone by modulating the catalase enzyme” was submitted to Antibiotics. It described the in vitro activity of β-lapachone to inhibit tested bacteria and their biofilms. After considering the paper, I suggest rejection with the following comments.

Response: I understand the reason why the reviewer is not happy with the presentation of the manuscript. I (the corresponding author) am extremely sorry. I do not know why the reviewer could not see the figures, tables, and their legends, though I uploaded all of them. Now I hope the reviewer will be able to see everything in the revised manuscript.

Overall comment

  1. English correction is necessary.

Response: English correction has been made

  1. 2. No result of figure and table in the paper. No legend of supplement figure.

Response: it will be there, I promise.

  1. No statistical analysis to compare different samples in each experiment.

Response: We used Microsoft Excel to calculate standard deviations and average values. We are just claiming that the compound has antibacterial and antibiofilm activities, in addition to the inhibition the catalase. We rarely compared the activities of β -Lap between the strains. If reviewer is not happy, we will do so.

In abstract

  1. “minimal inhibitory concentration (MICs) and minimal bactericidal concentration (MBCs)”. It can be replaced with broth microdilution assay.

Response: I agree, the same change has been made in the abstract section.

  1. Give the full name of MABA

Response: MABA is replaced by full form microplate alamarBlue assay

  1. 6. Give the abbreviated name after the first full name. Ex. β-lapachone (β-lap) or bacterial names.

Response: only abbreviated names have been given in whole text except at the beginning.

  1. Some word corrections such as “Growth kinetics inhibitin”, “biofilm inhibition assays, Biofilm eradication assay”.

In introduction

  1. Ref 5 may not be suitable for this overall statement “Treatment of bacterial infections are getting more difficult as a result of …”

Response: two more references incorporated, one about emerging of multidrug resistance strains and new pathogens.

  1. 9. Figure and molecular weight of β-lapachone can help readers to understand the paper.

Response: β -Lap structure is given in Fig. 5.

  1. 10. What is its MDR clinical isolates? Full name of MDR.

Response: full name given in the main text as “multi drug resistant”

  1. beta lactams can be changed to “other beta-lactam antibiotic”

Response: the change has been incorporated

In method

  1. 12. Method 2.2; 1. Italic font of bacterial name. 2. Give the abbreviated name of TS. Ex. trypticase soy (TS) broth.

Response: These issues have resolved in whole manuscript.

  1. 13. Method 2.4; 1. It is not necessary to give the detail of column number of the microplate. 2. “200 µL of distilled water” Is it sterile water? 3. Did the negative control and samples include 200 µL of sterile water? 4. What is “5 mM stock were prepared in DMSO”?

Response: Yes, water is sterile. Negative control are the wells containing sterile media and DMSO, while positive control wells contain culture and DMSO. Samples include culture wells having compound.

The working stock solution ranging from 10 mM- 0.00064 mM were prepared by 5-fold dilutions. In addition, a 5 mM stock was also prepared from 10 mM stock by two-fold dilutions and was included in the MIC determination. Now the statement has been edited in the text as “Working stocks of 10 mM, 5 mM, 2 mM, 0.4 mM, 0.08 mM, 0.0032 mM, and 0.00064 were prepared in DMSO”. We would like to keep the column numbers as it becomes easy to perform the assay.

  1. Method 2.5; What is agar used to grow the cells?

Response: Changed the heading to “Minimum bactericidal concentration by streaking method”.

  1. 15. Method 2.6; 1. The advantage of MABA can be removed. 2. The positive control is 20 mL DMSO + culture (media + cells). Is it negative control? 3. 20 mL DMSO + media is the blank for subtraction. 4. Change “water” to “sterile water”.

Response: the advantage of MABA has been removed. Yes, the reviewer is right, and it is explained in comment # 13. The Media + DMSO is used as blank for subtraction.

  1. Method 2.7; Did it have an empty disc containing β-lap?

Response: No,

  1. 17. Method 2.8; 1. Full name of MBIC. 2. Unit of 0.0016.

Response: Minimum biofilm inhibition concentration (MBIC), 0.0016 mM. The modifications have been incorporated in main manuscript text.

  1. 18. Method 2.9; 1. Change the reference of https://doi.org/10.1093/nar/gkac394 and listed it in the reference section.

Response: The doi has been removed and the reference has been incorporated in the method and reference sections of main manuscript file

In result

  1. 19. Result 3.1; 1. Italic font of bacterial names. 2. No data of figure and table, I cannot give comment.

Response: italics taken care off. Sorry for not having figure and table in the manuscript, though the same were uploaded.

  1. 20. Result 3.2; 1. There were 4 sensitive strains that β-lap inhibited. Why only three tested strains had a decrease in growth? 2. MIC or 1x MIC was used frequently. The authors could use one pattern for consistency. 3. The compound decreased the bacterial growth. This result should be related to its kinetics.

Response: We just took only three strains, there was no reason behind this.we are sure it would be same for the remianing strain too. Sure, the consistency has been maintained using MIC throughout the whole document

  1. Result 3.3; 1. The amount of compound or 20 mL of 2.5 mM β-lap should replace “2.5 mM β-lap”. 2. What is the effect of the combination of β-lap and antibiotics? (addition or synergy). 3. Docking parameters should be provided. 4. Catalase proximal active site for β-lap should be provided. 5. Fig or Figure, choose one pattern for consistency. 6. No explanation of Supplement Fig 2.

Response: Consistency of Fig. has been maintained throughout the document. 20 ul of 2.5 mM β-lap has been added to antibiotic discs. The antibacteial activity of only some antibiotics is increased in presence of β-lap. We cannot say it is synergic because we do not know the inhibition zone for β-lap alone as we didn’t have sterile discs. Figure. has been replaced by Fig.The explantion of Supl. Fig. 2 is given in its figure legend.Docking parameters are given in table 3, under column heading of center and docking size. For catalase proximal active site for β-lap are given in table 3 under column heading contact residues and His-55, Tyr-338 in Fig.5.

  1. Result 3.4; 1. What is “including 0.5 mM of β –Lap”? 2.How to calculate the inhibitory percentage of biofilm? 3. Give the definition of MBIC and why 0.04 mM was MBIC? 4. What is MBIC of P.aeruginosa?

Response: the statement is rewriten to clear the confusion. ---- the  cells were incubated with five indidual concentrations (0.5 mM, 0.2 mM, 0.04 mM, 0.008 mM and 0.0016 mM) of β –Lap for 24 hrs at 37oC. A formulla has been used to calculate the inhibitory percentage of biofilm and the statement about it has been made in material and methods section under crstal violet biofilm assay as“ the percentage of biofilm inhibition was calculated by the following formulla: Biofilm inhibition (%) = [(control OD570 nm – treated OD570nm)/control OD570nm] x 100. MBIC is the concentration, on which there is no inhibition of growth, but inhibition of biofilm formation. An ideal antibiofilm agent should not affect the growth of bacteria. Therefore a statement has been made in the text as MBIC can be defined as the minimum concentration of b-lap that exhibits  >50% of biofilm inhibition without affecting growth. Biofilm formation was not inhibited in P. aeruginosa under the conditions tested.

In discussion

  1. 23. “Out of ten bacterial strains screened”. Why it was 10 as 11 strains were tested in the method and result.

Response: a typo, it is 11 not 10. The typo has been fixed in main text of manuscrript. 

  1. To determine the synergy of β-lap and antibiotics, The checkerboard assay should perform to calculate the fractional inhibitory concentration (FIC) index in order to discuss the consequence of this combination.

Response: This was our preliminary observation regarding with the effect of compound on commercial antibiotics. We have not claimed about synergy or additive or indifferent effect of b-lap on any antibiotic. We have simply saw that b-lap increased the activity of antibiotics. I agree with the reviewer that checkboard assay would be ideal to comment on consequence of the combination, that is our future goal for sure.

  1. 25. More discussion on kinetic rate, IC50 on biofilm, and interaction between tested compound and catalase will improve the paper quality.

Response: The discussion part has been significantly improved.  

In conclusion

  1. 26. It is overstated that β-lap could inhibit at transcriptional level.

Response: the statement has been modified by removing the term transcriptional level.

Reviewer 3 Report

-          The manuscript should’ve been prepared better for review. The lines should’ve numbered in order to follow the possible error or concerns. I will convert the file in Word version in order to do that, but next time please keep in mind that this is not the way to send a manuscript to review.

-          At line 20 you should’ve written:”Growth kinetics inhibition in the presence of…”

-          At line 21, you should’ve written the ATCC code for every bacteria line

-          At line 22, you need to provide the meaning of the acronym in order for the reader to understand

-          At line 23, the correct writing would’ve been like this:” β -Lapachone was tested….”, Then write another sentence starting:” The compound showed potent antimicrobial activity against….”

-          At line 27, the correct writing would’ve been like this:” …in a dose dependent manner and in association with…”

-          At line 49 and 50, the correct writing would’ve been like this:“or to improve the efficacy of already  available antibiotics is the antimicrobial resistance (AMR)…”

-          At line 56, the plant species should be written in italics

-          At line 61, you should write the name of the disease with an uppercase

-          At line 63, write the name of the bacteria with an uppercase. Also, write in vitro in italics.

-          At line 66, please write the name of the acronym MDR, for those readers that are unfamiliar with acronym

-          At line 67, please write the name of the acronym TB, for those readers that are unfamiliar with acronym

-          At line 73, the correct writing would’ve been like this:” persistend cells…”

-          From line 96 to 99, you should write the name of the bacterial strain in italics

-          At line 106, explain the acronym TSA

-          At line 113, it is h for hour not hr

-          At line 153 and 154, please use a more accurate software to calculate or do statistics. Microsoft Excel has enormous errors

-          From line 221 to 236, you should write the name of the bacterial strain in italics

-          At line 262, “impregnated” is not a good word to describe your antibiotic placement. You should use “added to the discs that contained the conventional antibiotics”.

-          I see no figures in this article, although they are mentioned in the manuscript.

-          In the suplementari there are only a couple of the figures mentioned in the article, which need a good description under them. But the other figures I can’t find

-          You also need newer references, they are from too long ago.

Author Response

We are extremely thankful to the reviewer for his efforts and time that he/she spent while going through the manuscript. We appreciate his/her valuable comments, and we hope that he/she is satisfied with the responses.

The manuscript should’ve been prepared better for review. The lines should’ve numbered in order to follow the possible error or concerns. I will convert the file in Word version in order to do that, but next time please keep in mind that this is not the way to send a manuscript to review.

Response: Sorry for the inconvenience, the revised file would be number lined

-          At line 20 you should’ve written:”Growth kinetics inhibition in the presence of…”

Response: Here it is under methods section, if it would have been under result section then I believe it was appropriate.

-          At line 21, you should’ve written the ATCC code for every bacteria line

Response: others are laboratory strains

-          At line 22, you need to provide the meaning of the acronym in order for the reader to understand

Response: The full form of all acronyms has been provided

-          At line 23, the correct writing would’ve been like this:” β -Lapachone was tested….”, Then write another sentence starting:” The compound showed potent antimicrobial activity against….”

Response: it could be mentioned, but I believe the statement itself confirms it has been tested for antibacterial activity against the mentioned strains. In my opinion it would be a repetition and would increase the word limit of the abstract.

-          At line 27, the correct writing would’ve been like this:” …in a dose dependent manner and in association with…”

Response:  we didn’t use multiple concentrations of commercial antibiotics.

-          At line 49 and 50, the correct writing would’ve been like this:“or to improve the efficacy of already  available antibiotics is the antimicrobial resistance (AMR)…”

Response: I agree, it has been fixed.

-          At line 56, the plant species should be written in italics

Response: it has been italicized.

-          At line 61, you should write the name of the disease with an uppercase

Response:  it has been edited

-          At line 63, write the name of the bacteria with an uppercase. Also, write in vitro in italics.

Response: I agree, changes have been made

-          At line 66, please write the name of the acronym MDR, for those readers that are unfamiliar with acronym

Response: full form has been written

-          At line 67, please write the name of the acronym TB, for those readers that are unfamiliar with acronym

Response: full form has been given

-          At line 73, the correct writing would’ve been like this:” persistend cells…”

Response: changed to persistent cells

-          From line 96 to 99, you should write the name of the bacterial strain in italics

Response:

-          At line 106, explain the acronym TSA

Response: it is explained

-          At line 113, it is h for hour not hr

Response: replaced by hour

-          At line 153 and 154, please use a more accurate software to calculate or do statistics. Microsoft Excel has enormous errors

Response: the reviewer is right in his opinion, but we have only small data sets. I believe Microsoft Excel will suffice the need.

-          From line 221 to 236, you should write the name of the bacterial strain in italics.

Response: I agree, consistency has been maintained

-          At line 262, “impregnated” is not a good word to describe your antibiotic placement. You should use “added to the discs that contained the conventional antibiotics”.

Response: replaced with saturated

-          I see no figures in this article, although they are mentioned in the manuscript.

Response: I am extremely sorry for inconvenience, though all figures, tables, legends, and supplementary files were uploaded.

-          In the suplementari there are only a couple of the figures mentioned in the article, which need a good description under them. But the other figures I can’t find

Response: The full description has been given in legends

-          You also need newer references, they are from too long ago.

Response: some new references have been added

Reviewer 4 Report

Authors describe experiment on the activity of agains bacteria. They state "Molecular modelling predicted the inhibition of catalase activity by binding to
the proximal active site and heme binding sites of the enzyme" and this is further elaborated in the conclusion section but the results on this matter are scarce. I strongl recomment additional experiments. Moreover authors state that "
clinically important microorganisms by inhibiting the catalase either at its
activity or transcriptional level" but their study provides evidence only on several strains. To give this conclusion it is of importance to add more bacteria, but to also include other microorganisms (yeast, viruses, etc), or to clarify this conclusion and keep it at the level of their research.

The paper lacks some tables (use of control antibiotics, synergistic effects....).

Author Response

We are extremely thankful to the reviewer for his/her efforts and time that he/she spent while going through the manuscript. We appreciate his/her valuable comments and suggestions.  We hope that the responses are to his/her satisfaction level.

Authors describe experiment on the activity of agains bacteria. They state "Molecular modelling predicted the inhibition of catalase activity

by binding to the proximal active site and heme binding sites of the enzyme" and this is further elaborated in the conclusion section but the

results on this matter are scarce. I strongl recomment additional experiments. Moreover authors state that " clinically important

microorganisms by inhibiting the catalase either at its activity or transcriptional level" but their study provides evidence only on several

strains. To give this conclusion it is of importance to add more bacteria, but to also include other microorganisms (yeast, viruses, etc), or to

clarify this conclusion and keep it at the level of their research.

The paper lacks some tables (use of control antibiotics, synergistic effects....).

Response: our future aim is to study the type of inhibition (competitive, noncompetitive, or uncompetitive) of catalase by β –Lap in detail. I

agree to modify the conclusion as “Taken together, our data suggest that β –Lap is a promising novel antibacterial agent that inhibits the

bacterial catalase.” The revised manuscript has all figures, legends, and tables

Round 2

Reviewer 2 Report

Comment on the Antibacterial and antibiofilm activities of β –Lapachone by modulating the catalase enzyme.

I would like to thank the authors for answering my comment. I suggest the paper requiring minor revision before acceptance.

1. Fig 5B and 6; The color of alkyl, Pi-alkyl, and Pi-Pi stack interactions should be different for better clarification.

2. The legend of Fig 5C should include “Alignments of the conserved ligand binding sites …”

3. The italic font of bacterial name is necessary in Method 2.2 and Result 3.1.

4. The decimal number of inhibition zone against S. aureus in Table 1 must be the same as other value.

5. Supplementary Fig. 1; 1. Statistical analysis such as Student’s t-test to compare the significant difference will improve the work quality. 2. MIC can change to 1X MIC for consistency.

6. Fig 2, The author can replace DMSO with untreated samples. Otherwise, give the final concentration of DMSO used.

7. Fig 2, The author measured the growth kinetic for 16 h. You should show the growth of each tested stain until 16 h. (S. aureus 9 h and P. aeruginosa 12 h)

8. Table 2, 1. Statistical analysis such as Student’s t-test to compare the significant difference will improve the work quality. 2. Can it include SD of inhibition zone 3. Footnote may confuse the reader. You can choose one (- = no effect, R = resistance)

9. Fig 7, 1. Statistical analysis to compare the significant difference will improve the work quality. 2. It will be clear if you specified bar = biofilm inhibition and line = growth curve.

10. Fig 7, MBIC is defined as the concentration of β –Lap to give more than > 50% inhibition without affecting growth. Why MBIC of β –Lap in S. aureus is not 0.0016 mM?

11. Fig 8A and 8C, Statistical analysis to compare the significant difference will improve the work quality. If MBIC of β –Lap in S. aureus change based on item 10, the result of the mature biofilm disruption activity has to revise.

Author Response

We thank reviewer for his precious time that he spent in going through the manuscript and most importantly his valued suggestions and comments were very helpful. The comments have been addressed and accordingly the appropriate changes have been made in main text file of the manuscript, wherever required. For statistical analysis, we can mention the p values in main text. We found the p value for almost all samples less than 0.05. The values are like, 0.001, 0.003, 0.002, 0.006, and so on but all are less than 0.05. I would prefer to mention them in text or in the legend rather than giving in the figures.

Comment on the Antibacterial and antibiofilm activities of β –Lapachone by modulating the catalase enzyme.

I would like to thank the authors for answering my comment. I suggest the paper requiring minor revision before acceptance.

  1. 1. Fig 5B and 6; The color of alkyl, Pi-alkyl, and Pi-Pi stack interactions should be different for better clarification.

Response: we believe it is ok, if reviewer is not satisfied, we can change it.

  1. The legend of Fig 5C should include “Alignments of the conserved ligand binding sites …”

Response: It has been incorporated into the legend of the figure

  1. The italic font of bacterial name is necessary in Method 2.2 and Result 3.1.

Response: The changes have been made

  1. The decimal numberof inhibition zone against S. aureus in Table 1 must be the same as other value.

Response: It has been incorporated

  1. Supplementary Fig. 1; 1. Statistical analysis such as Student’s t-test to compare the significant difference will improve the work quality. 2. MIC can change to 1X MIC for consistency.

Response: We kept it as MIC to maintain consistency, as suggested by other reviewer also.

  1. Fig 2, the author can replace DMSO with untreated samples. Otherwise, give the final concentration of DMSO used.

Response: the concentration of DMSO is given in material and method’s section under subtitle “Microplate alamarBlue assay (MABA)” which is 10%

  1. Fig 2, The author measured the growth kinetic for 16 h. You should show the growth of each tested stain until 16 h. (S. aureus 9 h and P. aeruginosa 12 h).

Response: if you go beyond the time, for example beyond 9 h in S. aureus, the growth curve of control (DMSO treated) shows decline, the decline is not due to death of cells, it is due to reduction in the level of fluorescence of alamabrblue. I would like to re-frame the sentence to avoid confusion. It will be written as; “The growth at regular intervals of 30 min was monitored in FLUOstar® Omega microplate reader” Instead of “The growth at regular intervals of 30 min was monitored in FLUOstar® Omega microplate reader over the time of 16 hours”.

  1. Table 2, 1. Statistical analysis such as Student’s t-test to compare the significant difference will improve the work quality. 2. Can it include SD of inhibition zone 3. Footnote may confuse the reader. You can choose one (- = no effect, R = resistance)

Response: Foot note is corrected, only R= resistance retained. SD included in table.

  1. Fig 7, 1. Statistical analysis to compare the significant difference will improve the work quality. 2. It will be clear if you specified bar = biofilm inhibition and line = growth curve.

Response: Growth curve and bar have specified in legend of the figure.

  1. Fig 7, MBIC is defined as the concentration of β –Lap to give more than > 50% inhibition without affecting growth. Why MBIC of β –Lap in S. aureus is not 0.0016 mM?

Response: we will change the definition of the MBIC as “MBIC is defined the minimum concentration of β –Lap that exhibits highest biofilm inhibition without affecting growth” same has been incorporated in the main text file.

  1. Fig 8A and 8C, Statistical analysis to compare the significant difference will improve the work quality. If MBIC of β –Lap in S. aureus change based on item 10, the result of the mature biofilm disruption activity has to revise.

Response: Statement about MBIC changed as mentioned in previous comment’s (number 10) response.

Reviewer 4 Report

How did you determined the laboratory strains. I strongly recommend to use, especially for the first experiments to use characterised strains.If you have deterimend strains with no doubt using for example MS, please state. And again, more strains should be included in study. If you present this as preliminary study, perhaps you can hold and add more experiments. Use also appropriate method to determine the effect at transcriptional level, if this is actually true.

Author Response

We thank reviewer for his valuable comments and suggestions. All the comments have been addressed and the appropriate changes have been incorporated in the main text file, wherever required.

Comments and Suggestions for Authors. How did you determined the laboratory strains, I strongly recommend to use, especially for the first experiments to use characterised strains. If you have determined strains with no doubt using for example MS, please state. And again, more strains should be included in study. If you present this as preliminary study, perhaps you can hold and add more experiments. Use also appropriate method to determine the effect at transcriptional level, if this is actually true.

Response: the lab strains are being used in teaching practical to undergraduate and graduate students in the medical laboratory science program at the College of Applied Medical Science. These strains are being confirmed by many biochemical tests and morphological examinations. So there is no doubt about their characterization.

In the beginning we tested 11 bacterial strains, out of which only four showed zone of inhibition, others couldn’t. As there are only four strains, which showed susceptibility to b-lap, we changed the generalized statement of inhibition of catalase of bacteria to inhibition of catalase of four tested strains. We remained confined to only the susceptible strains tested in this study. B-lap might inhibit the growth (show zone of inhibition) of remaining strains at its higher concentration

In this study we just claim that the inhibition of catalase could be the cause of antibacterial activity of b-lap against the strains tested. We propose that there could be other reason as well in addition to oxidative stress, like transcriptional and translation regulation of catalase, which we don’t know and claim in the present study. But for sure in future we are going to look for transcriptional regulation of catalase in presence of b-lap (other strains and higher concentration of b-lap will be included), in addition to its mode of inhibition of catalase enzyme (either competitive or noncompetitive, or uncompetitive). We will use real time PCR to determine the mRNA level of katA in b-Lap-treated cells. We haven’t seen any article so far where the transcriptional regulation of catalase has been investigated in presence of b-lap. It will be interesting to investigate it.

Therefore, we would like to publish this part of work and in future come up full article describing the effect of b-lap on catalase at molecular level.

Round 3

Reviewer 4 Report

There is need to strenghten claims by using appropriate statistical analysis to compare the differences

Author Response

We thank reviewer for keenly going through the manuscript. His comments have been addressed and we agree with his valued suggestion.

Comments and Suggestions for Authors

Comment: There is need to strengthen claims by using appropriate statistical analysis to compare the differences

Response: we agree to his suggestion. We used IBM SPSS statistical software version 21 to perform the statistical analysis of data. One way ANOVA with Tukey’s post-hoc test was used to assess differences between treated and untried samples. The p < 0.05 was considered statistically significant. The ‘p’ values have been given in the legend of figures and corresponding symbols were inserted on top of columns.

Statement about statistical analysis have been incorporated in the materials and methods section of main manuscript file. In addition, the figure legends of both supplementary and main figures have been revised.